# Selective CO2 Fixation to Styrene Oxide by Ta-Substitution of Lindqvist-Type $[(Ta,Nb)_6O_{19}]^{8-}$ Clusters

Vorakit Chudatemiya [1], Mio Tsukada [1], Hiroki Nagakari [1], Soichi Kikkawa [1,2], Jun Hirayama [1,2], Naoki Nakatani [1], Takafumi Yamamoto [3] and Seiji Yamazoe [1,2,4,*]

[1] Department of Chemistry, Graduate School of Science, Tokyo Metropolitan University, 1-1 Minami-Osawa, Hachioji 192-0397, Tokyo, Japan
[2] Elements Strategy Initiative for Catalysts & Batteries (ESICB), Kyoto University, 1-30 Goryo-Ohara, Nishikyo-ku, Kyoto 615-8245, Japan
[3] Laboratory for Materials and Structures, Tokyo Institute of Technology, 4259 Nagatsuta-cho, Midori-ku, Yokohama 226-8503, Kanagawa, Japan
[4] Precursory Research for Embryonic Science and Technology (PRESTO), Japan Science and Technology Agency (JST), Tokyo 102-0076, Japan
* Correspondence: yamazoe@tmu.ac.jp

**Abstract:** Metal oxide clusters composed of group 5 metal ions, such as Nb and Ta, exhibit catalytic activities for CO2 fixation to styrene oxide (**SO**) due to the highly negative natural bonding charge of the terminal O atoms that could work as CO2 activation sites. In this study, tetrabutylammonium (TBA) salts of $[Ta_xNb_{6-x}O_{19}]^{8-}$ (TBA-Ta$_x$Nb$_{6-x}$, $x = 0$–6) were prepared and Ta-substitution effect on the catalytic properties of TBA-Ta$_x$Nb$_{6-x}$ for CO2 fixation to **SO** was investigated. We found that TBA-Ta$_1$Nb$_5$ shows the highest styrene carbonate (**SC**) selectivity (95%) among TBA-Ta$_x$Nb$_{6-x}$, although the **SO** conversion monotonously increases with the incremental Ta substitution amount. The CO2 fixation to **SO** under various conditions and in situ X-ray absorption fine structure measurements reveal that CO2 is activated on both terminal O sites coordinated to the Ta (terminal O$_{Ta}$) and Nb (terminal O$_{Nb}$) sites, whereas the activation of **SO** proceeds on the terminal O$_{Ta}$ and/or bridge O sites that are connected to Ta. Density functional theory (DFT) calculations reveal that the terminal O$_{Ta}$ of TBA-Ta$_1$Nb$_5$ preferentially adsorbs CO2 compared with other O$_{Nb}$ base sites. We conclude that the selective CO2 activation at terminal O$_{Ta}$ of TBA-Ta$_1$Nb$_5$ without **SO** activation is a crucial factor for high **SC** selectivity in the CO2 fixation to **SO**.

**Keywords:** mixed metal oxide clusters; CO2 fixation; base catalyst; polyoxometalate; Nb; Ta

## 1. Introduction

Several CO2 usage approaches have been implemented to reduce the impact of emitted greenhouse gases and to achieve a carbon neutral level. CO2 utilization in chemical production gained significant attention in replacing traditionally used C1 sources such as phosgene or CO, which post more toxicity. However, the activation of CO2 is challenging due to its thermodynamic stability, which requires appropriate catalysts, such as solid base catalysts, for instance, alkaline earth metal oxides. One of the useful reactions is CO2 cycloaddition to epoxides, which form cyclic carbonates. The resultant cyclic carbonates could be applied as solvents [1–5], monomers [6–9], electrolytes [10–13], and pharmaceuticals [14,15].

Recent applications of metal oxide clusters, namely, polyoxometalates, as base catalysts have been reported [16–22]. One of the advantages of utilizing metal oxide clusters over bulk solid base catalysts is that it does not require the surface activation of catalysts [23]. Up to now, the basicity of metal oxide clusters depends on the structures and the type of metal ions. The Lindqvist-type polyoxotungstate $[W_6O_{19}]^{2-}$ shows the basicity with p$K_a$ value of 11.1 and defective Keggin-type Ge-incorporated polyoxotungstate $[\gamma-H_2GeW_{10}O_{36}]^{6-}$

exhibits high basicity with a $pK_a$ value of 21.9 [24]. The series of group 5 metal polyoxometalates exhibit superior basicity to those of group 6 metal polyoxometalates, and the basicities of $[Nb_{10}O_{28}]^{6-}$, $[Nb_6O_{19}]^{8-}$, and $[Ta_6O_{19}]^{8-}$ increase to a $pK_a$ value of 23.8 [17,18]. Recently, Uchida's group reported that porous ionic crystals containing Nb/Ta were applied to Knoevenagel condensation reactions as base catalyst [19]. Density functional theory (DFT) calculations reveal that the base strength of the clusters is related to the natural bond orbital (NBO) charges of the surface O atoms and the higher negativity of the NBO charges leads to stronger basicity [16,18]. We reported that Lindqvist-type polyoxometalates with group 5 metal ions (Nb, Ta) had higher negative NBO charges compared to group 6 metal ions (Mo, W) and $[Nb_6O_{19}]^{8-}$ and $[Ta_6O_{19}]^{8-}$ could activate $CO_2$ and worked as catalysts for $CO_2$ fixation and conversion reactions [18,25]. The $CO_2$ was activated on the terminal O sites of metal oxide clusters, which are Lewis base sites, and activated $CO_2$ reacts with epoxides to form carbonates [17,18,21]. The cycloaddition of $CO_2$ to epichlorohydrin proceeded on Keggin-type $Na_{16}[SiNb_{12}O_{40}]$ [21]. In the case of Lindqvist-type $[M_6O_{19}]^{8-}$ (M = Nb, Ta), $[Ta_6O_{19}]^{8-}$ showed higher activity for $CO_2$ fixation to styrene oxide (**SO**) at 403 K than $[Nb_6O_{19}]^{8-}$ [18]. However, the styrene carbonate (**SC**) selectivity of $[Ta_6O_{19}]^{8-}$ was lower than 90% and byproducts were formed. In our previous study, Brønsted basicity was investigated using sodium salts of $[Ta_xNb_{6-x}O_{19}]^{8-}$ as solid base catalyst in Knoevenagel condensation reactions and local symmetry of $NbO_6$ and $TaO_6$ units in the clusters affected base catalytic properties [23]. In this study, tetrabutylammonium (TBA) salts of mixed metal oxide clusters $[Ta_xNb_{6-x}O_{19}]^{8-}$ (TBA-$Ta_xNb_{6-x}$, $x$ = 0–6) were prepared and applied to $CO_2$ fixation to **SO** to elucidate the Ta-substitution effect on the catalytic activities and selectivity. It was found that single-Ta-substituted TBA-$Ta_1Nb_5$ exhibited the highest **SC** selectivity among TBA-$Ta_xNb_{6-x}$. We demonstrated that the high **SC** selectivity was achieved by the selective adsorption of $CO_2$ on the terminal $O_{Ta}$ without **SO** activation under reaction conditions.

## 2. Results

The fabricated TBA-$Ta_xNb_{6-x}$ were characterized by X-ray absorption spectroscopy (XAS), electrospray ionization mass spectrometry (ESI–MS), Fourier-transformed infrared (FT-IR) in attenuated total reflectance (ATR) mode, and elemental analysis (Figures 1, S1 and S2, Table S1, respectively). ESI–MS suggests that the various components of Ta–Nb mixed metal oxide clusters are contained in the TBA-$Ta_xNb_{6-x}$. Ta $L_3$-edge Fourier-transformed extended X-ray absorption fine structure (FT-EXAFS) of TBA-$Ta_xNb_{6-x}$ indicates that peaks of Ta–M (M = Nb or Ta) shift to a longer length with increasing Ta content (Figure 1a). A similar peak shift of Nb–M (M = Nb or Ta) is observed in the Nb K-edge FT-EXAFS spectra (Figure 1b). Those indicate the Ta-substitution to Nb sites in $[Ta_xNb_{6-x}O_{19}]^{8-}$. Elemental analysis reveals that TBA/$[Ta_xNb_{6-x}O_{19}]^{8-}$ ratio is 5~6.

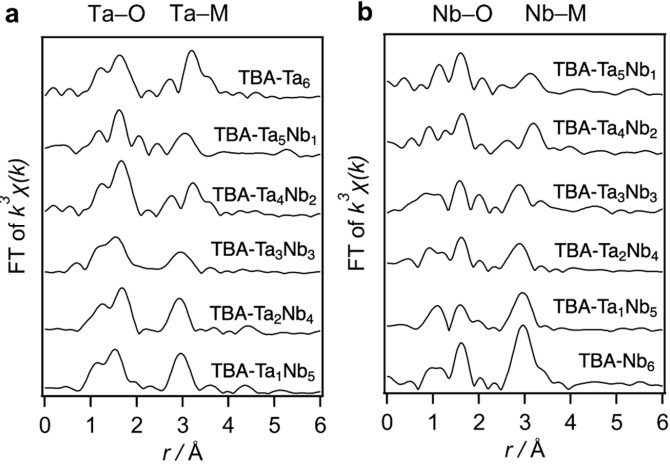

**Figure 1.** (**a**) Ta $L_3$-edge and (**b**) Nb K-edge FT-EXAFS spectra of TBA-$Ta_xNb_{6-x}$. The Ta–O (Nb–O) and Ta–M (Nb–M) are shown in $r$ ranges of 1.2–2.0 Å and 2.5–3.4 Å, respectively.

The prepared clusters were employed in the catalytic $CO_2$ fixation to **SO**. Figure 2 shows the **SO** conversion and **SC** selectivity for TBA-$Ta_xNb_{6-x}$ catalysts. The **SO** conversion gradually increases with incremental addition of Ta content and TBA-$Ta_6$ exhibits the highest **SO** conversion among them. On the other hand, the trend of **SC** selectivity for the composition of clusters differs from that of **SO** conversation (Figure 2). Interestingly, single-Ta-substituted TBA-$Ta_1Nb_5$ provides the highest **SC** selectivity (95%) among the mixed metal oxide clusters. Further Ta-substitution decreases **SC** selectivity and the **SC** selectivity becomes constant in Ta-rich TBA-$Ta_xNb_{6-x}$ catalysts ($x$ = 4–6). The number of TBA counteractions has a negligible impact on this reaction, although TBA/$[Ta_xNb_{6-x}O_{19}]^{8-}$ ratio varies with the composition [17]. The byproducts in this reaction over TBA-$Ta_6$ are polymers derived from the polymerization of **SO,** because the **SO** conversion is found for TBA-$Ta_6$ at 100 °C under $N_2$ atmosphere without $CO_2$ despite negligible **SO** conversion for TBA-$Nb_6$, as shown in Figure S3. The high **SC** selectivity (95%) of TBA-$Ta_1Nb_5$ maintains a high **SO** conversion (88%) for a 24 h reaction (Figure 2). In addition, the >95% SC selectivity is only achieved by TBA-$Ta_1Nb_5$ among the TBA-$Ta_xNb_{6-x}$ catalysts at >80% conversion (Figure S4). Thus, the selective $CO_2$ fixation to **SO** is achieved by the single Ta substitution to TBA-$Nb_6$.

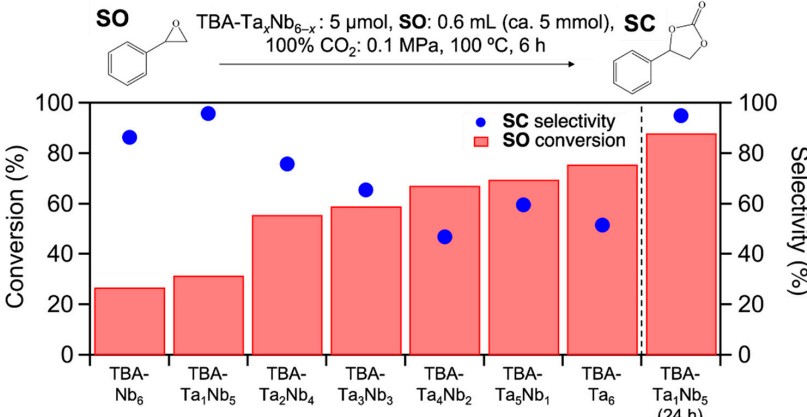

**Figure 2.** Results of $CO_2$ fixation to styrene oxide over TBA-$Ta_xNb_{6-x}$. Bars represent conversion of **SO**, blue dots represent selectivity of **SC**. Reaction condition: catalyst loading = 5 μmol, **SO** = 0.6 mL (ca. 5 mmol), 100% $CO_2$ = 0.1 MPa, temperature = 100 °C, reaction time = 6 h.

The time courses of $CO_2$ fixation to **SO** over TBA-$Ta_6$, TBA-$Ta_1Nb_5$, and TBA-$Nb_6$ are shown in Figures 3 and S5. The trends of conversion and selectivity depend on the composition of the clusters. TBA-$Ta_6$ exhibits the highest reaction rate among them, and **SC** selectivity increases with reaction time. TBA-$Ta_1Nb_5$ and TBA-$Nb_6$ show high **SC** selectivity at the initial stage of the reaction and the **SC** selectivity is maintained at a high **SO** conversion. Thus, TBA-$Nb_6$ and single-Ta-substituted TBA-$Ta_1Nb_5$ have the specific active sites for selective **SC** formation. The increment in the **SC** selectivity over TBA-$Ta_6$ is due to the consumption of **SO** and suppression of undesired reactions during the reaction.

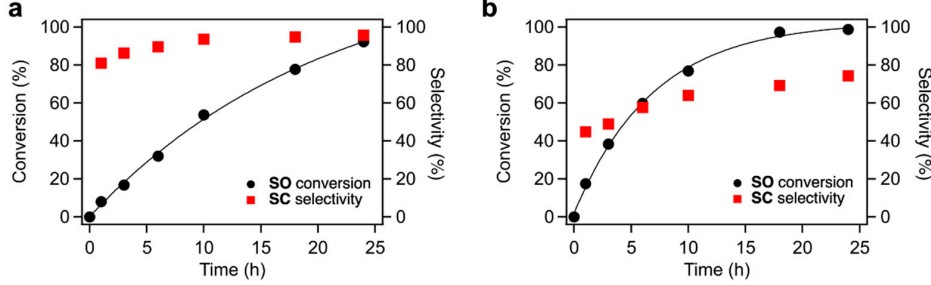

**Figure 3.** Time courses of **SO** conversion in $CO_2$ fixation to **SO** using (**a**) TBA-$Ta_1Nb_5$ and (**b**) TBA-$Ta_6$ as the catalysts. Reaction condition: catalyst loading = 10 μmol, **SO** = 1.2 mL (ca. 10 mmol), 100% $CO_2$ = 0.1 MPa, temperature = 100 °C.

Effects of **SO** concentration and $CO_2$ concentration on the $CO_2$ fixation to **SO** were studied for TBA-$Ta_1Nb_5$ (Figure 4). When increasing the **SO** concentration in the dimethyl sulfoxide (DMSO) solution, the **SC** selectivity slightly increases while maintaining the **SO** conversion. On the other hand, the **SC** selectivity dramatically decreases with decreasing $CO_2$ concentration (Figure 4b), suggesting the $CO_2$ activation is a key step in the $CO_2$ fixation to **SO**. We reported that $CO_2$ fixation to **SO** proceeds due to the fact that the $CO_2$ is activated on the terminal O sites and activated $CO_2$ reacts with **SO** to form **SC** [17,18]. The rate determining step of $CO_2$ fixation to **SO** is the nucleophilic attack of activated $CO_2$ to **SO**. The above results could be explained by the reaction mechanism. The decrease in the **SC** yield by reducing **SO** concentration, as shown in Figure 4a, is due to the inhibition of the reaction of activated $CO_2$ with **SO** by a low **SO** concentration. There are two reasons for the drastic decrease in **SC** selectivity by reducing $CO_2$ concentration. One is the decrease in the amount of activated $CO_2$. The other is that **SO** can be activated on TBA-$Ta_1Nb_5$ in low $CO_2$ concentration conditions. In fact, the **SO** conversion of TBA-$Ta_1Nb_5$ under $N_2$ atmosphere without $CO_2$ is higher than that of TBA-$Nb_6$, as shown in Figure S3, suggesting that the **SO** activation occurs by single Ta-substitution at a low $CO_2$ concentration.

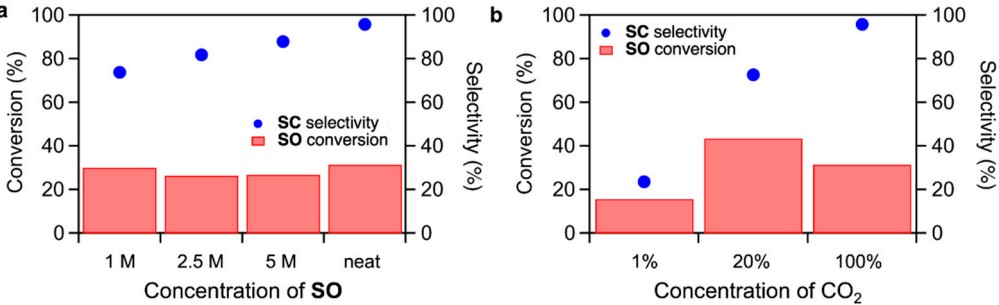

**Figure 4.** Effect of (**a**) **SO** concentration and (**b**) $CO_2$ concentration with $N_2$ balance. Reaction conditions: (**a**) TBA-$Ta_1Nb_5$ = 5 μmol, DMSO = 1 mL, **SO** = 0.6 mL (ca. 5 mmol for neat condition), 100% $CO_2$ = 0.1 MPa, temperature = 100 °C, reaction time = 6 h; (**b**) TBA-$Ta_1Nb_5$ = 5 μmol, **SO** = 0.6 mL (ca. 5 mmol), neat, $CO_2$ (0.1 MPa), temperature = 100 °C, reaction time = 6 h.

## 3. Discussion

We reported that the $CO_2$ fixation to **SO** proceeded on the terminal O sites of TBA-$Ta_6$ because the terminal O sites, which have the negatively charged O, work as Lewis base sites [18]. The $CO_2$ is activated on the terminal O sites and the activated $CO_2$ reacts with **SO** to form **SC**. The $CO_2$ adsorption on TBA-$Nb_6$, TBA-$Ta_1Nb_5$, and TBA-$Ta_6$ in **SO** was examined by in situ XAFS measurements (Figure 5). Nb K-edge XANES spectrum of TBA-$Nb_6$ in **SO** exhibits a pre-edge peak at 18,980.5 eV assigned to electron excitation from $1s$ to hybridized $4d-5p$ [26,27]. This pre-edge peak intensity gives us the information on the distortion from $NbO_6$ octahedral (*Oh*) symmetry. The pre-edge peak intensity decreases with the $CO_2$ introduction to TBA-$Nb_6$ in **SO**, which indicates that the *Oh* symmetry of $NbO_6$ in TBA-$Nb_6$ is improved by the $CO_2$ addition. Similar results were obtained for TBA-$Ta_1Nb_5$, as shown in Figure 5b. Ta $L_1$-edge XANES spectra indicate that the pre-edge peak at 11,686 eV, which is assigned to electron transition from Ta $2s$ orbitals to hybridized $5d-6p$ orbitals [26,27], decreases with the introduction of $CO_2$. This change of pre-edge peak intensity reveals that $TaO_6$ *Oh* symmetry is also improved by the $CO_2$ addition to TBA-$Ta_6$ and TBA-$Ta_1Nb_5$. These results suggest that the *Oh* symmetry of $TaO_6$ unit increases, while $NbO_6$ symmetry is slightly improved in TBA-$Ta_1Nb_5$ by $CO_2$ adsorption. Actually, the optimized structure of $[Ta_1Nb_5O_{19}]^{8-}$ with $CO_2$ adsorbed on the terminal $O_{Ta}$ site has highly *Oh* symmetric $TaO_6$ units compared to the bare $[Ta_1Nb_5O_{19}]^{8-}$ (Figure S6) $CO_2$ is also adsorbed on TBA-$Ta_xNb_{6-x}$ in **SO** solution. This structural change induced by $CO_2$ adsorption is also observed in FT-IR (Figure S7). FT-IR spectra of TBA-$Ta_1Nb_5$ in DMSO solvent show the characteristic absorption band assignable to the stretching vibration between the metal and terminal O atoms in $MO_6$ units (M=O bond). The absorption band

shifts to high energy, owing to the slight shrink in the O=Nb bond in $NbO_6$ units. Those results indicate that $CO_2$ is preferentially adsorbed on terminal O sites of $TaO_6$ unit and induces the structure change in TBA-$Ta_xNb_{6-x}$.

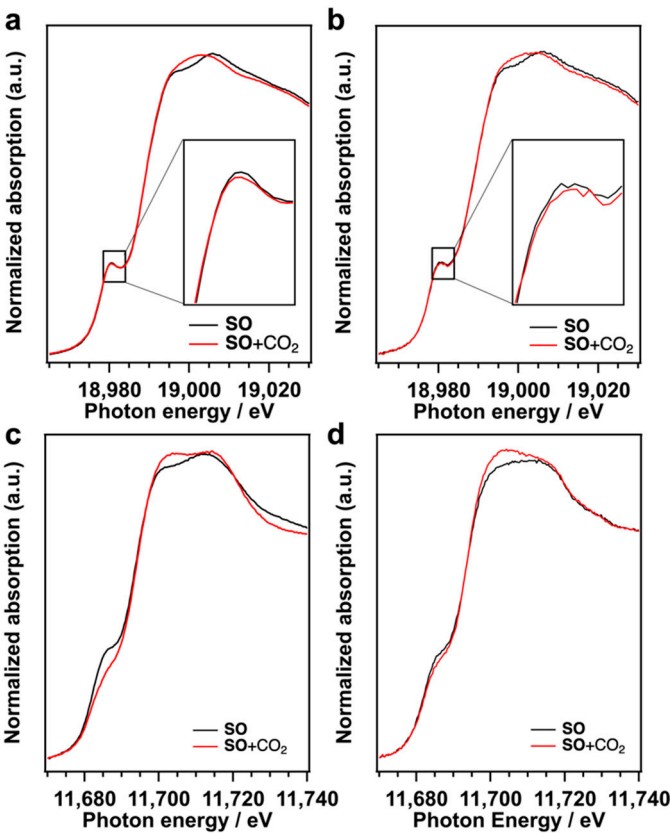

**Figure 5.** (**a**) Nb K-edge of TBA-$Nb_6$ and (**b**) TBA-$Ta_1Nb_5$, and (**c**) Ta $L_1$-edge XANES spectra of TBA-$Ta_6$ and (**d**) TBA-$Ta_1Nb_5$ in **SO** solution before (black line) and after (red line) introduction of $CO_2$.

Next, NBO charges of the surface O atoms of TBA-$Ta_xNb_{6-x}$ were calculated to elucidate the catalytic activity and selectivity of TBA-$Ta_xNb_{6-x}$ for $CO_2$ fixation to **SO** (Figure 6). The average NBO charge was also evaluated in Figure 6. The terminal O atoms coordinated to Ta (terminal $O_{Ta}$) have the most negative NBO charges (highest basicity) followed by the bridge O atom between Ta ions (bridge $O_{Ta-Ta}$) in TBA-$Ta_xNb_{6-x}$. The values of NBO charge of terminal $O_{Ta}$ hardly change with the Ta content. On the other hand, the NBO charge of the terminal O connecting to Nb (terminal $O_{Nb}$) has lower negativity than that of terminal $O_{Ta}$. The order of NBO charges in TBA-$Ta_xNb_{6-x}$ is terminal $O_{Ta}$, bridge $O_{Ta-Ta}$, bridge $O_{Ta-Nb}$ (Ta–O–Nb), terminal $O_{Nb}$, and bridge $O_{Nb-Nb}$ (Nb–O–Nb). As a result, the average NBO charges of TBA-$Ta_xNb_{6-x}$ gradually increase with increasing the Ta content. The catalytic activities of TBA-$Ta_xNb_{6-x}$, which increase with incremental addition of Ta content in Figure 2, could be explained by the average NBO charges, indicating the increase in the active sites of terminal $O_{Ta}$ by Ta substitution.

Finally, the $CO_2$ adsorption sites of $[Ta_1Nb_5O_{19}]^{8-}$ were also predicted by DFT calculations. We reported that $CO_2$ was preferentially adsorbed on terminal O sites rather than bridge O sites [18]. $[Ta_1Nb_5O_{19}]^{8-}$ has three terminal O sites (see Figure 7). To determine the $CO_2$ activation sites of $[Ta_1Nb_5O_{19}]^{8-}$, the $CO_2$ adsorption energy was calculated using three possible configurations (Figure 7). Among the three structures, the lowest energy is found in a structure with $CO_2$ adsorbed on the terminal $O_{Ta}$ site, which has the highest negative NBO charge among the surface oxygen atoms in $[Ta_1Nb_5O_{19}]^{8-}$. This result indicates that the $CO_2$ is preferentially adsorbed and activated on the terminal $O_{Ta}$. The **SO** adsorption energy was also calculated to gain the insight into **SO** activation sites (Figure S8). The

adsorption energy of **SO** on $O_{Ta}$ is lower than that of $CO_2$ on $O_{Ta}$ in $[Ta_1Nb_5O_{19}]^{8-}$. In addition, adsorption energies reveal that **SO** is more likely to be activated on $O_{Ta}$ than $O_{Nb}$. These results suggest that $CO_2$ preferentially adsorbs on $O_{Ta}$ site and it is unlikely that **SO** activation occurs on $O_{Nb}$ in TBA-$Ta_1Nb_5$. Therefore, high **SC** selectivity is achieved in TBA-$Ta_1Nb_5$. The low **SC** selectivity in Ta-rich TBA-$Ta_xNb_{6-x}$ is explained that not only by the fact that $CO_2$ but also **SO** is activated on $O_{Ta}$ sites by competitive adsorption due to the large number of $O_{Ta}$ adsorption sites.

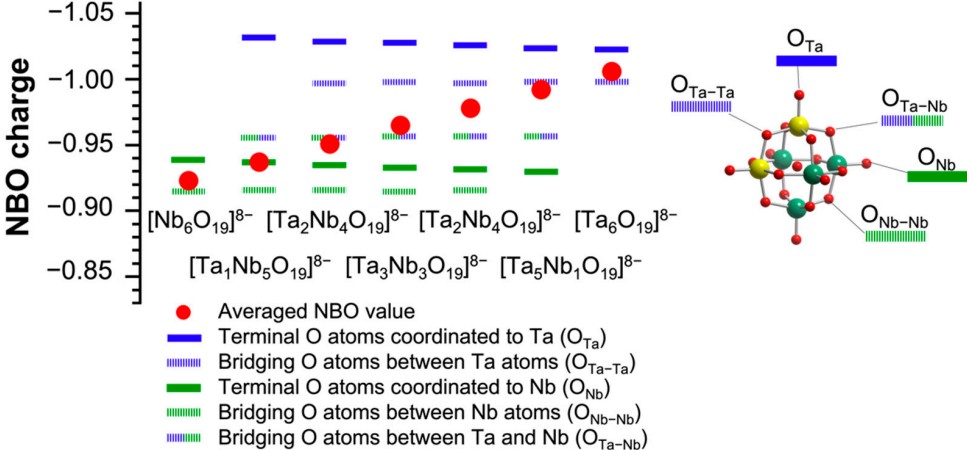

**Figure 6.** NBO charge of surface O sites of $[Ta_xNb_{6-x}O_{19}]^{8-}$. Color codes: yellow atom, Ta; green atom, Nb; blue line, Ta-coordinated O atoms; green line, Nb-coordinated O atoms; red, total O atoms. Red circle represents the averaged value. Terminal and bridging O atoms represent solid and dashed lines, respectively.

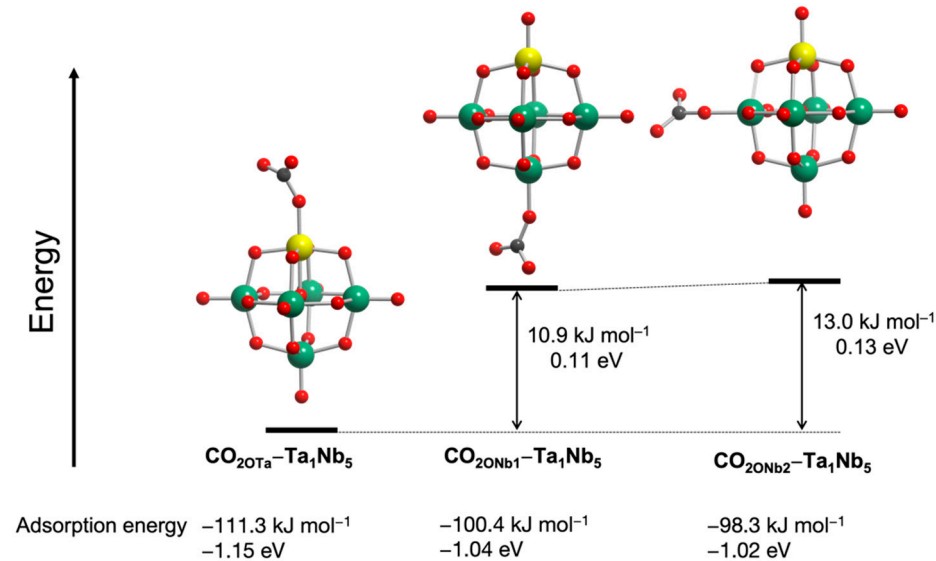

**Figure 7.** Total energy of $CO_2$-adsorbed $[Ta_1Nb_5O_{19}]^{8-}$ at different base sites.

The reaction mechanism of TBA-$Ta_xNb_{6-x}$ for $CO_2$ fixation to **SO** is discussed. In the case of TBA-$Nb_6$, $CO_2$ is adsorbed on the terminal $O_{Nb}$ sites and the activated $CO_2$ reacts nucleophilically with **SO** to form **SC**. The low catalytic activity of TBA-$Nb_6$ for $CO_2$ fixation to **SO** is due to the weak Lewis base strength (low negativity in NBO charges) of terminal $O_{Nb}$ compared with terminal $O_{Ta}$ of other TBA-$Ta_xNb_{6-x}$. The **SO** activation hardly occurs on TBA-$Nb_6$, as shown in Figure S3, which is one of the reasons why TBA-$Nb_6$ shows high **SC** selectivity. The **SO** conversion gradually increases with Ta substitution amount, as shown in Figure 2. This can be explained by the increase in active terminal $O_{Ta}$ sites. On the other hand, the **SC** selectivity decreases for high Ta content of TBA-$Ta_xNb_{6-x}$

($x \geq 2$). The low **SC** selectivity is due to the **SO** activation on the surface of TBA-Ta$_x$Nb$_{6-x}$ ($x \geq 2$), as shown in Scheme 1. In fact, TBA-Ta$_6$ exhibits the highest **SO** conversion among TBA-Ta$_6$, TBA-Ta$_1$Nb$_5$, and TBA-Nb$_6$ in the absence of CO$_2$ accompanied with a viscosity increase (Figure S3). The sharp contrast in **SO** conversion in the absence of CO$_2$ conditions for TBA-Ta$_6$ and TBA-Nb$_6$ clearly indicates that terminal O$_{Ta}$ and/or bridge O$_{Ta-Ta}$ can activate **SO**. On the other hand, TBA-Ta$_1$Nb$_5$ exhibits the highest **SC** selectivity among TBA-Ta$_x$Nb$_{6-x}$ despite having a terminal O$_{Ta}$ site. The DFT calculation (Figure 7) and CO$_2$ concentration dependence on CO$_2$ fixation to **SO** (Figure 4b) reveal that the single terminal O$_{Ta}$ in [Ta$_1$Nb$_5$O$_{19}$]$^{8-}$ preferentially adsorbs CO$_2$ at 100% CO$_2$ conditions and **SO** is not activated on terminal O$_{Nb}$, bridge O$_{Nb-Nb}$, and bridge O$_{Ta-Nb}$ (Scheme 1). We conclude that the selective CO$_2$ activation at the terminal O$_{Ta}$ in TBA-Ta$_1$Nb$_5$ without **SO** activation is a crucial factor for high **SC** selectivity in the CO$_2$ fixation to **SO**.

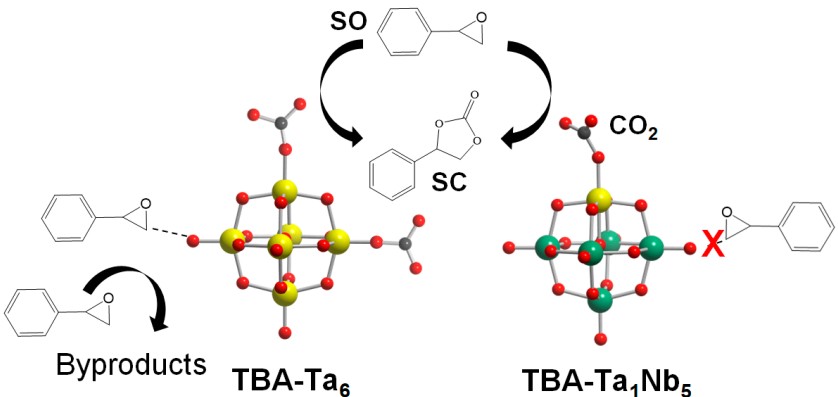

**Scheme 1.** Activation of CO$_2$ on O$_{Ta}$ leads to a reaction with **SO** to form **SC**, while adsorption of **SO** on O$_{Ta}$ leads to the formation of byproducts observed in reaction promoted by TBA-Ta$_6$. In the case of TBA-Ta$_1$Nb$_5$, **SO** is not activated on O$_{Nb}$ and preferential adsorption of CO$_2$ on O$_{Ta}$ results in high selectivity of **SC**.

## 4. Materials and Methods

TBA salts of [Ta$_x$Nb$_{6-x}$O$_{19}$]$^{8-}$ (TBA-Ta$_x$Nb$_{6-x}$, $x$ = 0–6) were prepared by microwave-assisted hydrothermal synthesis (Biotage Initiator$^+$ 400 W) using Ta$_{2(x/6)}$Nb$_{2(1-x/6)}$O$_5$·$n$H$_2$O as the precursors. First, Na$_3$Ta$_{x/6}$Nb$_{1-x/6}$O$_4$ were prepared by modified solid-state reaction method according to the reported procedures [23,28]. M$_2$O$_5$ (M = Ta or Nb), Na$_2$C$_2$O$_4$, and (NH$_2$)$_2$CO at a molar ratio between (Ta + Nb):Na:(NH$_2$)$_2$CO of 1:1:4 was ground to fine powder prior to calcination at 773 K for 4 h to obtain NaTa$_{x/6}$Nb$_{1-x/6}$O$_3$. NaTa$_{x/6}$Nb$_{1-x/6}$O$_3$ was mixed with Na$_2$C$_2$O$_4$, and (NH$_2$)$_2$CO at a molar ratio between NaTa$_{x/6}$Nb$_{1-x/6}$O$_3$:Na:(NH$_2$)$_2$CO of 1:1:3 followed by calcination at 1173 K for 4 h. The resulting powder, Na$_3$Ta$_{x/6}$Nb$_{1-x/6}$O$_4$, was characterized by XRD (Rigaku Miniflex) having diffraction patterns corresponding to the references (Figure S9). Na$_3$Ta$_{x/6}$Nb$_{1-x/6}$O$_4$ was dissolved in water and 1 M HCl was added until pH of the supernatant reached 1 or less. The white precipitate was collected by centrifugation and washed with pure water until the pH of the supernatant became neutral. After drying in vacuum and oven, the Ta$_{2(x/6)}$Nb$_{2(1-x/6)}$O$_5$·$n$H$_2$O was obtained. Then, 10% tetrabutylammonium hydroxide (TBAOH) aqueous solution was added to Ta$_{2(x/6)}$Nb$_{2(1-x/6)}$O$_5$·$n$H$_2$O. The mixture was reacted using microwave-assisted hydrothermal synthesis at 180 °C for 5−15 min. The resultant product was washed with hexane to obtain TBA$_6$H$_2$[Ta$_x$Nb$_{6-x}$O$_{19}$]. The fabricated clusters were characterized by ESI–MS (Figure S1) (Bruker, MicroOTOFII-ST1), Fourier-transformed infrared spectrometry (JASCO, FT/IR-4700) equipped with attenuated total reflectance-infrared spectroscopy (JASCO, ATR-PRO ONE) (Figure S2), elemental analysis (Table S1), and X-ray absorption fine structure (XAFS) analysis (BL01B1, SPring-8) (Figure 1). XAFS spectra were recorded in transmittance mode using ionization chambers as detectors at room temperature. Si(111) double-crystal monochromator was used to

obtain the incident X-ray beam for Ta $L_1$- and $L_3$-edges XAFS. In the case of Nb K-edge XAFS measurements, Si(311) double-crystal monochromator was employed. The data were analyzed using xTunes software [29]. The XANES spectra were extracted as the extended X-ray absorption fine structure (EXAFS) after normalization at edge height. The EXAFS spectra in the *k* range 3.0–14.0 Å$^{-1}$ were Fourier-transformed into *r* space to obtain FT-EXAFS spectra. The illustrations of $Ta_xNb_{6-x}O_{19}$ were computed using VESTA [30].

In general, $CO_2$ fixation to **SO** over TBA-$Ta_xNb_{6-x}$ were carried out using 5 μmol of catalyst, **SO** (0.6 mL, ca. 5 mmol), 100% $CO_2$ (0.1 MPa) at 100 °C for 6 h using biphenyl as an internal standard. The product solutions were analyzed using gas chromatography equipped with flame ionization detector (GC-FID, Shimadzu, GC-2014 with column Restex, Rtx-1) and gas chromatography–mass spectrometry (GC–MS, Shimadzu, GCMS-QP2010 SE with column Agilent, DB-1MS). Time course of $CO_2$ fixation to **SO** reactions were carried out using 10 μmol of catalyst, **SO** (1.2 mL, ca. 10 mmol), 100% $CO_2$ (0.1 MPa) at 100 °C for 24 h. Small amount of solution (ca. 20 μL) was drawn to measure at specified reaction times. The peak areas from GC-FID chromatograms were used to calculate with this formula:

$$Conversion\ (\%) = \left(1 - \frac{Area_{Sub.}}{Area_{IS}} \times \frac{Area_{IS0}}{Area_{Sub.0}}\right) \times 100$$

$$Selectivity(\%) = \frac{Area_{Pro.}}{Area_{IS}} \times \frac{ECN_{Sub.}}{ECN_{Pro.}} \times \frac{Area_{IS0}}{Area_{Sub.0}} \times \frac{100}{Conversion\ (\%)} \times 100$$

where *Sub.* = substrate (**SO**), *IS* = internal standard (biphenyl), *Pro.* = product (**SC**), *ECN* = equivalent carbon number, subscripted 0 = initial value before reaction.

The DFT calculations were conducted using Gaussian 16 program as previously reported [18]. The structural optimization for $[Ta_xNb_{6-x}O_{19}]^{8-}$ was performed by B3LYP with the solvation effect of DMSO using PCM (dielectric constant = 46.826). LanL2DZ basis sets were employed for Ta and Nb atoms and $6-31 + G(d)$ basis sets for O and C atoms to investigate the effect of the composition of the clusters on the NBO charge of O atoms and the adsorption energies of $CO_2$ on $[Ta_1Nb_5O_{19}]^{8-}$.

## 5. Conclusions

In conclusion, TBA-$Ta_xNb_{6-x}$ (*x* = 0–6) were prepared by microwave-assisted hydrothermal reaction and were used as catalysts for $CO_2$ fixation to **SO** to produce **SC**. Among TBA-$Ta_xNb_{6-x}$, TBA-$Ta_1Nb_5$ shows the highest selectivity toward **SC,** whereas the **SO** conversion increases with the Ta content in the clusters. The effects of **SO** concentration and $CO_2$ concentration for $CO_2$ fixation to **SO** indicate that high **SO** concentration and 100% $CO_2$ atmosphere are required to obtain high **SC** selectivity for TBA-$Ta_1Nb_5$ because the rate-determining step of $CO_2$ fixation to **SO** is the reaction of the activated $CO_2$ with **SO**. The **SO** conversions in the absence of $CO_2$ suggest that the **SO** activation hardly occurs in TBA-$Nb_6$. DFT calculations reveal that the increase in the **SO** conversion with Ta content is the increment of the active terminal $O_{Ta}$ sites that have the highest negative NBO charges. In addition, $[Ta_1Nb_5O_{19}]^{8-}$ preferentially adsorbs $CO_2$ at terminal $O_{Ta}$ sites compared to other $O_{Nb}$ sites. In conclusion, the selective $CO_2$ activation at terminal $O_{Ta}$ in TBA-$Ta_1Nb_5$ without **SO** activation is a crucial factor for high **SC** selectivity in the $CO_2$ fixation to **SO**.

**Supplementary Materials:** The following supporting information can be downloaded at: https://www.mdpi.com/article/10.3390/catal13020442/s1, Table S1: Elemental analysis results of the synthesized $TBA_6H_2[Ta_xNb_{6-x}O_{19}]$; Figure S1: ESI–MS (negative ion mode) spectra of $TBA_mH_n[Ta_xNb_{6-x}O_{19}]^{8-m-n}$ measured in aqueous solutions; Figure S2: FT-IR spectra (ATR mode) of (a) TBA-$Ta_6$, (b) TBA-$Ta_5Nb_1$, (c) TBA-$Ta_4Nb_2$, (d) TBA-$Ta_3Nb_3$, (e) TBA-$Ta_2Nb_4$, (f) TBA-$Ta_1Nb_5$, and (g) TBA-$Nb_6$; Figure S2: FT-IR spectra (ATR mode); Figure S3: $CO_2$ fixation to **SO** promoted by TBA-$Ta_xNb_{6-x}$ under $N_2$ atmosphere under $N_2$ atmosphere; Figure S4: Results of $CO_2$ fixation to **SO** over TBA-$Ta_xNb_{6-x}$; Figure S5: Time course of $CO_2$ fixation to **SO** and selectivity of **SC** over TBA-$Nb_6$; Figure S6: Optimized structure of $[Ta_1Nb_5O_{19}]^{8-}$ and $CO_2$-adsorbed $[Ta_1Nb_5O_{19}]^{8-}$; Figure S7: In situ FT-IR spectra (ATR mode) of TBA-$Ta_1Nb_5$ in DMSO before (black line) and after $CO_2$ adsorption

(red line). (b) Optimized structure of $[Ta_1Nb_5O_{19}]^{8-}$ and $CO_2$-adsorbed $[Ta_1Nb_5O_{19}]^{8-}$. Figure S8: Total energy of **SO**-adsorbed $[Ta_1Nb_5O_{19}]^{8-}$ at different base sites. Figure S9: XRD patterns of $Na_3Ta_{x/6}Nb_{6-x/6}O_4$.

**Author Contributions:** S.K. and S.Y. designed this study. V.C., M.T. and J.H. synthesized and characterized catalysts and carried out the catalytic tests. H.N. and N.N. carried out the DFT calculations and analyzed the NBO charges of the catalysts. T.Y. measured and analyzed the XRD. V.C., S.K., M.T. and S.Y. measured and analyzed the XAFS spectra. All authors have read and agreed to the published version of the manuscript.

**Funding:** This study was financially supported by NEDO (JPNP14004), JSPS KAKENHI (No. 20K22467, 21H01718, and 22K14543), Tokyo Human Resources Fund for City Diplomacy, Tokyo Metropolitan University Research Fund for Young Scientists, Tokyo Metropolitan Government Advanced Research (R3-1), and Yazaki Memorial Foundation for Science and Technology.

**Data Availability Statement:** The data that support the findings of this study are available from the corresponding author upon reasonable request.

**Acknowledgments:** The synchrotron radiation experiment was performed at BL01B1 in SPring-8 under the approval of the Japan Synchrotron Radiation Research Institute (JASRI) as 2020A1068, 2021B1380, 2021A1406, 2021B1535, 2022A1532, 2022A1627, 2022B1684, and 2022B1911.

**Conflicts of Interest:** The authors declare no conflict of interest.

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
