# Peer review of "Selective CO2 Fixation to Styrene Oxide by Ta-Substitution of Lindqvist-Type [(Ta,Nb)6O19]8− Clusters"

_catalysts, doi:10.3390/catal13020442_

Round 1
Reviewer 1 Report
CO2 utilization in chemical production has gained significant attention. Fixation of CO2 to styrene oxide for generating the styrene carbonate is one of the useful reactions. In this work, the authors applied Ta-substituted Lindqvist-type [TaxNb6-xO19]8- as the catalyst for this reaction. The Ta1Nb5 showed the highest 95% selectivity, which was attributed to the selective CO2 activation at terminal OTa in Ta1Nb5 without styrene oxide activation. However, the effect of Ta content on the reaction mechanism is still elusive, and some essential characterization is lacking. Some detailed comments:
1. Page 3, line 84. Tetrabutylammonium (TBA) cations are commonly used as co-catalyst in this type of reaction, and their content in the reaction system is also a crucial factor to activate the CO2. However, the TBA contents in TaxNb6-x may affect the reaction performance. Such influence had not been taken into consideration both in the performance evaluation and mechanism part.
2. Page 3, Figure 2. Comparing the conversion between Nb6 and Ta1Nb6, only a minor increase was observed in 6 h. The 88% conversion of Ta1Nb6 was achieved in 24 h. The paper should compare all the catalytic data after the reaction was finished. As mentioned in Figure 3, the reaction rate of different catalysts are different, and the comparison of data at 6 h make nonsense.
3. Page 5, Figure 5. The XAFS results suggested that the TaO6 Oh symmetry is improved by the CO2 addition. What is the relationship between CO2 addition and symmetry improvement? The exact coordination model should be given out before and after CO2 addition.
4. Page 5, Line 138. The XAFS experiments were only exerted on Nb6 and Ta6, similar experiments were absent for the other TaxNb6-x.
5. It was claimed that the terminal OTa and/or bridge OTa−Ta can activate styrene oxide. Whereas the CO2 was also activated by the terminal OTa. Will there be competition between the two processes? Which molecule was favorable to combine with OTa. The DFT calculations towards the styrene oxide were suggested.
Reviewer 2 Report
The paper describes the use of Nb-Ta clusters for CO2 fixation to epoxide rings. The authors employed proper characterization techniques and the conclusions support the experimental findings. The work is highly novelty due to the importance of this reaction. Some observations that could improve the work are:
1. Please revise Figures 3b and S4. The caption is the same in both figures.
2. If the byproducts are related to polymers formed, ¿How is explained that with the reaction progress, the SC selectivity increased?
3. The authors summarize the results based on the charge of oxygen clusters as active sites, and they assume that CO2 adsorption is the rate-determining step (rds), proposing a mechanism (plausible) based on only a step of the reaction. ¿The authors performed some DFT calculus for the adsorption of SO? ¿Some other evidence kinetic could confirm this step as the rds?
4. The authors should explore the use of CO2-clusters (CC) or hybrid materials containing these CC as novel materials to this type of reactions.
Reviewer 3 Report
The authors report metal oxide clusters composed of Nb and Ta metal ions showing catalytic activities for CO2 fixation to styrene oxide. They found that SO conversion monotonously increased with the increment of Ta substitution, and the TBA-Ta1Nb5 exhibited the highest SC selectivity. The authors using XANES and DFT calculation to study the reaction mechanism. However, the authors’ conclusion can not be well confirmed according to the present analysis. The authors should address these issues before considering of publication in Catalysts.
(1) In page 3 line 91, the author said that the SC selectivity decreased with the increase in the Ta substitution amount. However, from Figure 2 the SC selectivity of TBA-Ta5Nb1 and TBA-Ta6 is higher than that of TBA-Ta4Nb2. How do the authors explain the discrepancy?
(2) In page 4 line 119-122, the TBA-Ta6 under N2 exhibited the highest SO conversion from Figure S3, suggesting the ability of SO activation over the TBA-Ta6. The DFT results indicated that the CO2 can be activated most easily over TBA-Ta6 due to the highest NBO charge. Thus, the TBA-Ta6 should show the highest activity and selectivity of SC. How does the TBA-Ta6 show lower selectivity?
(3) The authors should give reason why the SC selectivity of TBA-Ta6 increased with the increment of conversion. This may be helpful for understanding the mechanism of CO2 fixation to SO.
(4) According to the results of XANES and DFT, CO2 is easily adsorbed on the terminal OTa sites. However, these results can not provide enough CO2 activation information. The CO2 activation is usually investigated by the change of C-O vibration or banding energy (such as in-situ FTIR).
(5) The by-products of the CO2 fixation to SO should be provided.

Round 2
Reviewer 1 Report
The quality of this paper was improved and my concerns were solved. I think this paper can be accepted.
Author Response
We deeply thank the reviewer for the critical comments and useful suggestions that have helped us to improve our paper.
Reviewer 3 Report
All my questions have been addressed. I recommend this paper for publication in its present form.
Author Response
We are grateful to the reviewer for the useful suggestions that have helped us to improve our paper.